

# Intercomparison of aerosol extinction profiles retrieved from MAX-DOAS measurements

U. Frieß[1], H. Klein Baltink[2], S. Beirle[3], K. Clémer[4,b], F. Hendrick[4], B. Henzing[5], H. Irie[6], G. de Leeuw[5,7,8], A. Li[9], M. M. Moerman[5], M. van Roozendael[4], R. Shaiganfar[3], T. Wagner[3], Y. Wang[9,3], P. Xie[9], S. Yilmaz[1], and P. Zieger[10,a]

[1]Institute of Environmental Physics, University of Heidelberg, Heidelberg, Germany
[2]Royal Netherlands Meteorological Institute (KNMI), De Bilt, the Netherlands
[3]Max Planck Institute for Chemistry, Mainz, Germany
[4]BIRA-IASB, Brussels, Belgium
[5]Netherlands Organization for Applied Scientific Research (TNO), Utrecht, the Netherlands
[6]Center for Environmental Remote Sensing, Chiba University, Chiba, Japan
[7]Finnish Meteorological Institute (FMI), Helsinki, Finland
[8]Department of Physics, University of Helsinki, Helsinki, Finland
[9]Anhui Institute of Optics and Fine Mechanics, Chinese Academy of Sciences, Hefei, China
[10]Paul Scherrer Institute, Laboratory of Atmospheric Chemistry, Villigen, Switzerland
[a]now at: Stockholm University, Department of Environmental Science and Analytical Chemistry, Stockholm, Sweden

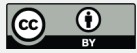

[b]now at: Instituut voor Sterrenkunde, Leuven University, Leuven, Belgium

Received: 19 November 2015 – Accepted: 9 December 2015 – Published: 15 January 2016

Correspondence to: U. Frieß (udo.friess@iup.uni-heidelberg.de)

Published by Copernicus Publications on behalf of the European Geosciences Union.

**AMTD**

doi:10.5194/amt-2015-358

**MAX-DOAS aerosol intercomparison**

U. Frieß et al.

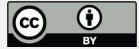

Discussion Paper | Discussion Paper | Discussion Paper | Discussion Paper |

**AMTD**

doi:10.5194/amt-2015-358

**MAX-DOAS aerosol intercomparison**

U. Frieß et al.

## Abstract

A first direct intercomparison of aerosol vertical profiles from Multi-Axis Differential Optical Absorption Spectroscopy (MAX-DOAS) observations, performed during the Cabauw Intercomparison Campaign of Nitrogen Dioxide measuring Instruments (CINDI) in summer 2009, is presented. Five out of 14 participants of the CINDI campaign reported aerosol extinction profiles and aerosol optical thickness (AOT) as deduced from observations of differential slant column densities of the oxygen collision complex ($O_4$) at different elevation angles. Aerosol vertical profiles and AOT are compared to backscatter profiles from a ceilometer instrument and to sun photometer measurements, respectively. Furthermore, the near-surface aerosol extinction coefficient is compared to in-situ measurements of a humidity controlled nephelometer and dry aerosol absorption measurements. The participants of this intercomparison exercise use different approaches for the retrieval of aerosol information, including the retrieval of the full vertical profile using optimal estimation and a parametrised approach with a prescribed profile shape. Despite these large conceptual differences, and also differences in the wavelength of the observed $O_4$ absorption band, good agreement in terms of the vertical structure of aerosols within the boundary layer is achieved between the aerosol extinction profiles retrieved by the different groups and the backscatter profiles observed by the ceilometer instrument. AOT from MAX-DOAS and sun photometer show a good correlation ($R > 0.8$), but all participants systematically underestimate the AOT. Substantial differences between the near-surface aerosol extinction from MAX-DOAS and from the humidified nephelometer remain largely unresolved.

## 1  Introduction

Aerosols play an important role in the atmospheric system. Aerosol particles scatter and absorb radiation, but also affect the formation, optical properties, and lifetime of clouds, and therefore have an impact on the radiation balance of the Earth's atmo-

Discussion Paper | Discussion Paper | Discussion Paper | Discussion Paper |

**AMTD**

doi:10.5194/amt-2015-358

sphere. However, the impact of aerosols on the climate system is still only poorly understood (Stocker et al., 2013). Direct emission of soot particles, as well the formation of secondary organic aerosols and the condensation of atmospheric gases on aerosol particles (e.g., sulfuric acid or organic vapours), affect air quality and human health.

Various chemical processes in the atmosphere can be strongly affected by aerosols, since these provide surfaces for heterogeneous reactions. Examples are the heterogeneous formation of nitrous acid on soot particles (Ammann et al., 1998), the autocatalytic release of reactive bromine on sea salt aerosols in Polar Regions (Simpson et al., 2007), and the stratospheric ozone depletion as a consequence of halogen activation

on polar stratospheric clouds (Crutzen and Arnold, 1986).

A quantification of the optical properties, spatial distribution and chemical composition of aerosols is crucial for an understanding of these processes. Therefore, measurement techniques for the determination of the amount, vertical distribution and optical properties of aerosols using a relatively simple and cost-effective instrumentation are

15 highly desirable. On the other hand, knowledge on the spatial distribution of aerosols and their impact on the radiative transfer is also important for the interpretation of passive atmospheric remote sensing observations from ground and satellite. The usage of Multi-Axis Differential Optical Absorption Spectroscopy (MAX-DOAS) measurements for the retrieval of atmospheric aerosol properties (Hönninger et al., 2004; Wagner

et al., 2004; Frieß et al., 2006), has found a growing number of applications during recent years (e.g., Irie et al., 2008, 2009; Lee et al., 2009; Takashima et al., 2009; Clémer et al., 2010; Li et al., 2010; Vlemmix et al., 2010; Zieger et al., 2011; Frieß et al., 2011; Wagner et al., 2011; Sinreich et al., 2013; Wang et al., 2014; Hendrick et al., 2014; Vlemmix et al., 2015).

As part of these studies, MAX-DOAS aerosol profiles, aerosol optical thickness (AOT) and/or surface extinction were compared to established instrumentation, such as lidar, sun photometer and in-situ aerosol instruments. These intercomparison studies are of great value for the validation of MAX-DOAS aerosol retrievals, but suffer from several difficulties. A comparison of the AOT from MAX-DOAS and sun photome-

**MAX-DOAS aerosol intercomparison**

U. Frieß et al.

Title Page

| Abstract | Introduction |
| Conclusions | References |
| Tables | Figures |

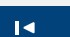 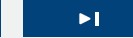

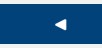 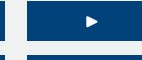

Back | Close

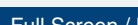

Interactive Discussion

ter does not allow for a validation of the retrieved profile shape. Compared to lidar, MAX-DOAS has a much coarser vertical resolution and a different altitude sensitivity. Backscatter lidar instruments only provide information on the backscatter signal, and a determination of the actual aerosol extinction from these measurements is subject

to large uncertainties. Therefore comparisons of backscatter lidar with MAX-DOAS extinction profiles can only be performed on a qualitative basis. Raman lidar systems can directly measure aerosol extinction profiles, but suffer from a low signal-to-noise ratio during daylight, while MAX-DOAS measurements cannot be performed at night. A further shortcoming of lidar measurements is the limited overlap between lidar beam and

FOV of the receiving telescope which leads to a lack of reliable data near the surface where MAX-DOAS is most sensitive. A comparison of MAX-DOAS measurements with in-situ instrumentation, such as nephelometer and Multi-Angle Absorption Photometer (MAAP) is complicated by the fact that in-situ instruments perform point-like measurements, usually directly at or near the surface, whereas the aerosol surface extinction

from MAX-DOAS represents an average over a certain height range with a typical vertical extent of 50–100 m. For this study, these complications are partly overcome by using a common aerosol inlet at 60 m above ground. The in-situ aerosol measurements are therefore expected to be more comparable to the MAX-DOAS observations than for an inlet directly at the surface. Most aerosol in-situ instruments measure quan-

tities which are not directly comparable to MAX-DOAS. Aerosols can take up water and therefore their optical properties – especially the particle light scattering coefficient – strongly depend on the ambient relative humidity (RH) (Zieger et al., 2013). Continuous ground-based measurements by nephelometer instruments are usually performed at dry conditions. Here a RH-controlled nephelometer is used to retrieve the ambient

value in addition to dry particle light absorption measurements (Fierz-Schmidhauser et al., 2010; Zieger et al., 2011). A general problem of comparisons between remote sensing and in-situ observations is that MAX-DOAS usually measures different air masses, with the retrieved aerosol profiles being representative for an average over

**AMTD**

doi:10.5194/amt-2015-358

**MAX-DOAS aerosol intercomparison**

U. Frieß et al.

Discussion Paper | Discussion Paper | Discussion Paper | Discussion Paper

the light paths in the lowermost troposphere that extend horizontally over several kilo-metres.

Here we present first direct intercomparisons of aerosol extinction profiles retrieved using MAX-DOAS measurements and aerosol retrieval algorithms from several work-groups. The measurements were performed in the framework of the Cabauw Intercom-parison Campaign of Nitrogen Dioxide measuring Instruments (CINDI) at the Cabauw Experimental Site for Atmospheric Research (CESAR) in the Netherlands (51.97° N, 4.93° E), during June/July 2009. An overview of the campaign as well as details of the instrumentation and DOAS data analysis can be found in Piters et al. (2012) and Roscoe et al. (2010). In total, 22 instruments from 14 institutes participated in the cam-paign, of which five participants delivered data on the aerosol vertical distribution or on AOT. During CINDI, MAX-DOAS measurements were performed continuously by all instruments in a westnorth-westerly direction (around 287° azimuth angle). The nom-inal set of elevation angles included 90, 30, 15, 8, 4, and 2°, but some instruments also observed skylight from additional directions. A primary objective of CINDI was the intercomparison of the differential slant column densities (dSCDs) of $NO_2$ and the oxygen collision complex $O_4$ measured by MAX-DOAS. A previous study has demon-strated that the $O_4$ dSCDs from the different instruments participating in the CINDI campaign, which serve as input for the aerosol retrieval algorithms, show good agree-ment (Roscoe et al., 2010). Therefore, a comparison of aerosol properties derived from the measured $O_4$ dSCDs allows to investigate differences in the various retrieval algo-rithms, which use a variety of different approaches, as well as the choice of different retrieval parameters (e.g., the a priori).

## 2 Retrieval of atmospheric aerosol properties from MAX-DOAS

MAX-DOAS measurements of scattered sunlight yield dSCDs, i.e. the difference $dS(\alpha) = S(\alpha) - S_{ref}$ between the slant column density of atmospheric trace gases mea-sured at an elevation angle $\alpha$ (angle between the horizon and the line of sight, LOS)

**AMTD**

doi:10.5194/amt-2015-358

**MAX-DOAS aerosol intercomparison**

U. Frieß et al.

Interactive Discussion

Discussion Paper | Discussion Paper | Discussion Paper | Discussion Paper |

and a reference measurement $S_{ref}$. For aerosol and trace gas retrievals, usually a zenith sky measurement of the same elevation sequence, i.e. closest in time to the off-axis measurements, is chosen as reference. The slant column density represents the integrated trace gas concentration along the light path, $S = \int \rho(s)\,ds$, with the integral representing the weighted average over individual light paths through the atmosphere. The oxygen collision complex $O_4$ exhibits pronounced absorption structures in the UV/Vis spectral region (Greenblatt et al., 1990). Since its concentration is proportional to the square of the $O_2$ concentration, which is well known, variations in the $O_4$ dSCDs are caused by variations in the atmospheric light path, which is altered by the presence of aerosols. Therefore measurements of the oxygen collision complex $O_4$ at different LOS allow for the retrieval of atmospheric aerosol properties. Alternatively, or in addition to the $O_4$ dSCDs, relative intensities, i.e. the ratio of the detector signal measured in the zenith and in off-axis directions, can be used to retrieve atmospheric aerosol properties (Frieß et al., 2006).

Since MAX-DOAS measurements only contain indirect information on the aerosol vertical profile, inverse methods are necessary for the retrieval procedure (Frieß et al., 2006). In general, aerosol properties are derived by comparing the measured $O_4$ dSCDs (and/or relative intensities) at different elevation angles to simulations from radiative transfer models (RTM). Using non-linear inversion algorithms, the aerosol properties that serve as input for the RTM are altered until best agreement between measurement and simulation is achieved. A general problem that MAX-DOAS has in common with other atmospheric remote sensing techniques is the limited information content of the measurements. As a consequence, the full state vector (e.g., an aerosol extinction profile $k(z)$ at high vertical resolution) cannot be reconstructed without any further constraints to the results. Here, different approaches are possible: either a Bayesian approach is applied where additional constraints are posed in the form of an a priori state vector, or a parametrisation with only a small number of quantities describing the aerosol vertical distribution (e.g., the aerosol optical thickness (AOT) or the layer height and AOT of a box profile) is used. The solution of the former approach

**AMTD**

doi:10.5194/amt-2015-358

**MAX-DOAS aerosol intercomparison**

U. Frieß et al.

Title Page

Abstract | Introduction

Conclusions | References

Tables | Figures

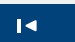 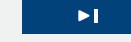

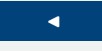 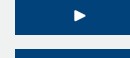

Discussion Paper | Discussion Paper | Discussion Paper | Discussion Paper |

is retrieved using the well-known optimal estimation method (OEM) (Rodgers, 2000), whereas the latter approach is based on more simple least squares methods (LSM). In general, the solution of the inverse problem $\hat{x}$ is determined by minimising a cost function in the form of

$$\chi^2 = (\boldsymbol{y} - \mathbf{F}(\boldsymbol{x}, \boldsymbol{b}))^\top \mathbf{S}_\varepsilon^{-1}(\boldsymbol{y} - \mathbf{F}(\boldsymbol{x}, \boldsymbol{b})) + (\boldsymbol{x} - \boldsymbol{x_a})^\top \mathbf{S}_\mathbf{a}^{-1}(\boldsymbol{x} - \boldsymbol{x_a}) \tag{1}$$

Here, $\mathbf{F}(\boldsymbol{x}, \boldsymbol{b})$ is a forward model (here: a radiative transfer model), which describes the measurement $\boldsymbol{y}$ (the $O_4$ dSCDs and/or relative intensities) as a function of the atmospheric state $\boldsymbol{x}$ (the aerosol vertical profile). The vector $\boldsymbol{b}$ represents additional forward model parameters (e.g. aerosol single scattering albedo and phase function) which are not retrieved. In case of OEM algorithms, the a priori state vector $\boldsymbol{x_a}$ with covariance $\mathbf{S_a}$ serves as an additional constraint, which has to be considered because the information content of the measurement is usually too low to allow for a full reconstruction of the atmospheric state on the basis of the measurements only. In case of LSM, the a priori information represented by the second term in Eq. (1) is omitted (i.e., $\mathbf{S_a}^{-1} \equiv 0$), and only a small number of parameters (i.e., layer height and AOT) is retrieved. The covariance matrix $\mathbf{S}_\varepsilon$ describes the uncertainties in the measurement (in case of LSM sometimes set to unity if no error weighting is performed). The vertical resolution of the retrieval is quantified by the so-called averaging kernel matrix $\mathbf{A} = \partial \hat{x}/\partial x$, which represents the sensitivity of the retrieved profile as a function of the true atmospheric profile. The retrieved profile $\hat{x}$ can be represented as the true profile $\boldsymbol{x}$, smoothed by the averaging kernel matrix $\mathbf{A}$ according to

$$\hat{x} = \boldsymbol{x_a} + \mathbf{A}(\boldsymbol{x} - \boldsymbol{x_a}) \tag{2}$$

The general features of the different algorithms participating in the intercomparison are summarised in Table 1, and the individual retrieval algorithms are briefly described in the following sections. $O_4$ dSCDs or relative intensities measured at elevation angles of 2, 4, 8, 15, 30°, relative to a zenith sky spectrum of the same sequence, serve as input measurement vector. Some participants (Heidelberg and JAMSTEC) do not

Discussion Paper | Discussion Paper | Discussion Paper | Discussion Paper | Discussion Paper |

**AMTD**

doi:10.5194/amt-2015-358

**MAX-DOAS aerosol intercomparison**

U. Frieß et al.

use single elevation sequences but all observations within a fixed time period (20 and 30 min, respectively) as input vector. All participants except the Max Planck Institute for Chemistry (MPIC) use OEM algorithms for the retrieval. MPIC uses an LSM algorithm for the retrieval of AOT and aerosol layer height (see Sect. 2.4). For the intercompari-

son, a reference wavelength of 477 nm has been chosen since most of the participants use the $O_4$ absorption band at this wavelength for the aerosol retrieval. Aerosol properties measured at other wavelengths (retrievals from MPIC, as well as ceilometer, sun photometer and humidified nephelometer) are converted to 477 nm using the Ångström coefficient $\alpha$ derived from co-located sun photometer measurements at wavelengths of

440 and 675 nm. In contrast to all other retrieval algorithms, the Anhui Institute of Optics and Fine Mechanics (AIOFM) uses observed relative intensities in addition to $O_4$ dSCDs as input vector (for details see Sect. 2.5). Furthermore, AIOFM did not participate in the CINDI campaign with own instruments, but use data measured by the Heidelberg instrument as input for their own retrieval algorithm.

The a priori profiles for the BIRA, Heidelberg, AIOFM and JAMSTEC retrievals are shown in Fig. 1. Heidelberg and AIOFM use similar a priori profiles with an aerosol extinction at the surface of 0.1 and 0.08 km$^{-1}$, respectively, and a linear decrease with altitude. The BIRA algorithm assumes a significantly smaller a priori aerosol extinction, with a surface value of 0.05 km$^{-1}$ and an exponential decrease with altitude. The JAM-

STEC algorithm represents the aerosol profile on a much coarser vertical grid than the other algorithms using three layers of 1 km thickness each, and assumes a larger a priori extinction with a value of 0.126 km$^{-1}$ in the lowermost layer. More specific information on the choice of the a priori profiles and the a priori covariance matrices can be found in the following sections. Depending on the information content of the mea-

surements, or more specifically the values of the averaging kernels in each layer, there will be a potential bias of the retrieved aerosol extinction profiles towards the a priori profiles (see Eq. 2). This influence of the a priori profile on the resulting extinction profiles needs to be considered when comparing the results from the different retrieval algorithms.

**AMTD**

doi:10.5194/amt-2015-358

**MAX-DOAS aerosol intercomparison**

U. Frieß et al.

Discussion Paper | Discussion Paper | Discussion Paper | Discussion Paper |

## 2.1 The BIRA retrieval algorithm

The BIRA-IASB OEM-based profiling tool called bePRO is extensively described in Clémer et al. (2010) and Hendrick et al. (2014). The forward model is the linearized discrete ordinate radiative transfer (LIDORT) model (Spurr, 2008). The LIDORT code includes an analytical calculation in a pseudo-spherical geometry of the weighting functions needed for the profile inversion. This allows for near real time automated retrievals of aerosol and trace gas vertical distributions without the use of pre-calculated look-up tables. The standard vertical grid implemented in bePRO consists of ten layers of 200 m thickness between 0 and 2 km, two layers of 500 m between 2 and 3 km and 1 layer between 3 and 4 km altitude. Pressure and temperature profiles are taken from the Air Force Geophysics Laboratory (AFGL) database. For each scan, the $O_4$ dDSDs measured at 477 nm at 10 elevation angles (1, 2, 3, 4, 5, 8, 10, 15, 30, and 90°) serve as the measurement vector $\boldsymbol{y}$. The corresponding $\mathbf{S}_\varepsilon$ matrix is constructed as a diagonal matrix, with variances equal to the square of the $O_4$ DOAS fitting error. An exponentially decreasing aerosol extinction profile with an AOT of 0.05 and a scaling height of 1 km is used as a priori. The a priori error covariance matrix is set as in Clémer et al. (2010): the diagonal element corresponding to the lowest layer, $\mathbf{S}_a(1,1)$, is equal to the square of a scaling factor $\beta$ ($\beta = 0.1$ in the present case) times the maximum partial AOT of the profile. The other diagonal elements decrease linearly with altitude down to $0.2 \cdot \mathbf{S}_a(1,1)$. The off-diagonal terms in $\mathbf{S}_a$ are set using Gaussian functions with a correlation length of 50 m. The aerosol single scattering albedo (SSA) and phase functions needed for the weighting functions calculations are derived off-line based on the co-located Aeronet sun photometer measurements. The surface albedo is fixed to 7 %.

## 2.2 The Heidelberg retrieval algorithm

The HeiPro retrieval is an updated version of the algorithm already described in detail in Frieß et al. (2006, 2011). It is based on the optimal estimation method and retrieves

**AMTD**

doi:10.5194/amt-2015-358

**MAX-DOAS aerosol intercomparison**

U. Frieß et al.

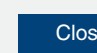
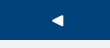
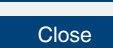
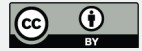

the most probable state vector by minimising the cost function given by Eq. (1). The radiative transfer model SCIATRAN (Rozanov et al., 2005a) serves as forward model for the retrieval. The state vector $x$ consists of the logarithm of the extinction in 20 layers of 200 m thickness, extending from the surface up to 4 km altitude. Using the logarithm of the extinction instead of the actual extinction has the advantage that negative values are avoided, which cannot be processed by the radiative transfer model. The $O_4$ dSCDs measured at 477 nm at elevation angles of 90, 30, 15, 8, 4, and 2° serve as the measurement vector $y$, and the diagonal values of the measurement covariance matrix $S_e$ are set to the square of the dSCD measurement errors. All measurements within a fixed time interval of 20 min serve as measurement vector. Given a measurement time of about 7 min for a single elevation sequence, this means that the measurement vector usually contains several measurements at the same elevation angle. The a priori has an extinction of 0.1 km$^{-1}$ at the surface, is linearly decreasing from the surface up to an altitude of 3.5 km and is constant above with a value of 0.0033 km$^{-1}$. This a priori profile has been smoothed with a 7 points running average. The a priori error (square root of the diagonal elements of $S_a$) has been set to 100 % of the a priori at all altitudes, and the non-diagonal elements of $S_a$ are exponentially decreasing with distance between layer altitudes with a correlation length of 1 km. For the radiative transfer calculations, aerosol single scattering albedo and asymmetry parameter $g$ were adapted from the co-located Aeronet sun photometer measurements. For this intercomparison, all profiles retrieved by the Heidelberg groups were used without any further quality filtering of the data.

## 2.3 The JAMSTEC retrieval algorithm

The Japanese MAX-DOAS profile retrieval algorithm version 1 (JM1) applied to observations performed by JAMSTEC is described in detail in Irie et al. (2011). It is based on the optimal estimation method to solve the nonlinear inversion problem. The state vector consists of AOT and three parameters determining the shape of the vertical profile. An advantage of this parameterization is that the absolute value of the aerosol

Discussion Paper | Discussion Paper | Discussion Paper | Discussion Paper |

**AMTD**

doi:10.5194/amt-2015-358

**MAX-DOAS aerosol intercomparison**

U. Frieß et al.

**AMTD**

doi:10.5194/amt-2015-358

**MAX-DOAS aerosol intercomparison**

U. Frieß et al.

extinction is unnecessary in the state vector. Instead, a priori knowledge of the profile shape is needed. The aerosol extinction is given as the product of the AOT and profile shape but the aerosol extinction retrieval is less subject to a prior knowledge of the AOT and profile shape as the resulting a priori error for the aerosol extinction is large. The adopted parameterization primarily yields partial AOT values or mean aerosol extinc­tion values for layers of 0–1, 1–2, 2–3, 3–100 km. Since a vertical profile shape within each layer is considered, extraction of aerosol extinction coefficients at any altitude is possible (Irie et al., 2008). A lookup table of the box-air-mass-factor vertical profile used in the forward model was created using the JACOSPAR radiative transfer model, which was developed based on its predecessor MCARaTS (the Monte Carlo Atmospheric Radiative Transfer Simulator, Iwabuchi, 2006). Parameters for the JAMSTEC retrieval were 0.95 for the single scattering albedo, 0.65 for the asymmetry parameter (under the Henyey–Greenstein approximation), and 0.1 for the surface albedo.

## 2.4 The MPIC retrieval algorithm

The MPIC profile inversion is described in detail in Wagner et al. (2011). It is based on the comparison of the measured $O_4$ absorption (analysed using the absorption bands at 360 and 380 nm in a joint fitting window ranging from 353 to 390 nm) with simulated $O_4$ Differential Airmass factors (DAMFs). The retrieved $O_4$ DSCDs are converted into DAMFs by dividing them by the atmospheric $O_4$ vertical column density (VCD). From vertical profiles of temperature and pressure at Cabauw the $O_4$ VCD was determined to $1.43 \times 10^{43}$ molec$^2$ cm$^{-5}$ (for the units, see Greenblatt et al., 1990). Finally, the $O_4$ DAMFs are scaled by a constant factor of 1.2 to account for possible systematic un­certainties of the $O_4$ cross section (see Wagner et al., 2009; Clémer et al., 2010). Our retrieval follows the method of Li et al. (2010) with slight modifications described in Wagner et al. (2011). Atmospheric $O_4$ DAMF are simulated using the radiative transfer model McARTIM (Deutschmann et al., 2011) assuming a large variety of atmospheric aerosol extinction profiles, which are described by a simple parameterisation scheme: for the CINDI campaign the total aerosol optical depth and the layer height were var-

Discussion Paper | Discussion Paper | Discussion Paper | Discussion Paper

**AMTD**

doi:10.5194/amt-2015-358

**MAX-DOAS aerosol intercomparison**

U. Frieß et al.

ied. The profile shape is composed of two parts: a box profile from the surface to the layer height with a constant aerosol extinction, and an exponentially decreasing part above (5 % of the total AOT are contained in this exponentially decreasing part). From a least-squares fitting procedure between the measured and simulated $O_4$ DAMF, the total aerosol optical depth and layer height of the box profile are determined for each elevation sequence. The aerosol extinction is derived by dividing the AOT of the box profile (95 % of the total AOT) by the layer height. The errors of the retrieved profiles are assessed based on (a) the residual sum of squares between the measurement and the model results and (b) from the fit process itself taking into account the sensitivity of the measured quantities with respect to variations of the profile parameters.

## 2.5 The AIOFM retrieval algorithm

The "Profile inversion algorithm of aerosol extinction and trace gas concentration developed at AIOFM in cooperation with MPIC" (PriAM) (Wang et al., 2013) is applied to the $O_4$ dSCDs and relative intensities from Heidelberg MAX-DOAS instrument to retrieve profiles of aerosol extinction. The PriAM algorithm is based on the optimal estimation method (Rodgers, 2000) and implements a nonlinear iterative approach which is based on the Gauss–Newton method modified by Levenberg–Marquardt to speed up the minimization of the cost function. The measurements vector $y$ consists of the $O_4$ dSCDs and relative intensities at 477 nm in each measurement sequence. The measured $O_4$ dSCDs are scaled down by a factor of 0.8. Including relative intensities is a strong constraint for the AOT and improve the sensitivity of the inversion on the upper layers (Frieß et al., 2006). The a priori profile $x_a$ is a linear decreasing profile with an AOT of 0.15. A priori uncertainty covariance matrix $S_a$ is non-diagonal with the diagonal elements of the square of 33 % of $x_a$ and non-diagonal elements calculated from the Gaussian function with the correlation length of 0.5 km (Frieß et al., 2006). A diagonal measurement uncertainty covariance matrix $S_\epsilon$ has the diagonal elements of the square of 100 % fitting errors of the $O_4$ dSCDs and 1.5 % of the relative intensities. Due to the deviation of the true aerosol phase function from the Henyey–Greenstein

parameterization (Henyey and Greenstein, 1941) used in the model simulations (Wang et al., 2015) for the forward scattering, artifacts occur in the retrieved aerosol profiles at small relative azimuth angles when including intensity. Considering this effect, the error of the intensity has been increased from 1.5 to 3 % in the afternoon. This effectively decreases the weight of the information from relative intensities compared to the information from $O_4$ dSCDs. For the measurements on 1 and 2 July, the relative intensity has been excluded from the retrieval for relative azimuth angles below 20°. The weighting function **K** is calculated using the full-spherical RTM SCIATRAN 2.2 (Rozanov et al., 2005b).

## 2.6  Complementary measurements

A large variety of aerosol measurements, both in-situ and by remote sensing, were performed during the CINDI campaign: backscatter and Raman lidar systems as well as a ceilometer measured the vertical distribution of aerosol in terms of backscatter and extinction profiles; two nephelometer systems, one of which was humidity controlled, as well as a multi-angle absorption photometer measured the scattering and absorption properties of aerosol particles; finally, a sun photometer measured the AOT.

Backscatter profiles measured by a Vaisala LD40 ceilometer regularly operated at the CESAR site by KNMI are used for the validation of the aerosol profiles retrieved from MAX-DOAS. The ceilometer has a vertical resolution of 30 m and measures backscatter profiles every 30 s at a wavelength of 905 nm from about 120 m up to 11.5 km altitude using a pulsed InGaAs laser diode. Due to the limited overlap between outgoing laser beam and the field of view of the collecting telescope, no valid backscatter data is available for altitudes below 120 m.

The AOT at 440, 675, 870 and 1020 nm, as well as the corresponding Ångström parameters, single scattering albedo and phase function, are retrieved from continuous measurements at the CESAR site by an automated CIMEL CE 318 Sun Photometer using direct sunlight measurements. This instrument, operated by TNO, is part of the AErosol RObotic NETwork (AERONET). A summary of the AERONET Level 2 data

**AMTD**

doi:10.5194/amt-2015-358

**MAX-DOAS aerosol intercomparison**

U. Frieß et al.

during the golden days of CINDI campaign (see Sect. 3) is shown in Fig. 2. The AOT varies between 0.1 and 0.7, with a mean and standard deviation of 0.34 and 0.18, respectively. The Ångström exponent, which describes the wavelength dependence of the aerosol extinction, amounts to $1.49 \pm 0.14$. The aerosol single scattering albedo during CINDI is significantly lower than at other urban sites (Dubovik et al., 2002), with values as low as 0.84 at the beginning of the campaign, and a mean value of $0.92 \pm 0.03$, indicating that significant amounts of absorbing particles are present. Furthermore, a mean asymmetry parameter of $0.72 \pm 0.02$ has been retrieved from Sun Photometer measurements.

The aerosol scattering coefficient $k_s$ near the surface was determined by a humidified nephelometer (WetNeph) in combination with a simultaneously operated dry air Nephelometer. The WetNeph is described in detail by Fierz-Schmidhauser et al. (2010), and a comparison of extinction coefficients from MAX-DOAS and WetNeph has already been described in Zieger et al. (2011). Briefly, the aerosol scattering coefficient $k_s$ as well as the back scattering coefficient $k_b$ are measured at three wavelengths (450, 550, and 700 nm) at defined relative humidities between 20 and 95 % using an integrating nephelometer (TSI Inc., Model 3563). The WetNeph measurements allow the determination of the ambient particle extinction coefficient, assuming that the particle absorption coefficient does not change with RH. The ambient particle extinction coefficient can then be directly compared to the retrieved value of the MAX-DOAS without any further assumption on particle growth in humid air. The ambient RH measurements were taken at six different locations on the 200 m high mast. The inlet of the WetNeph was located at a height of 60 m at the Cabauw tower.

# 3   Results

In this section, quantities derived from the different aerosol retrieval algorithms are validated against independent measurements. Aerosol extinction profiles are compared to ceilometer measurements in Sect. 3.1. The comparison of retrieved AOT and surface

**AMTD**

doi:10.5194/amt-2015-358

**MAX-DOAS aerosol intercomparison**

U. Frieß et al.

**AMTD**

doi:10.5194/amt-2015-358

**MAX-DOAS aerosol intercomparison**

U. Frieß et al.

extinction with data from sun photometer and in-situ aerosol observations, respectively, is discussed in Sect. 3.2. For the comparison, eight days with predominantly clear sky conditions ("golden days") were selected. These were 23–25 June, as well as 30 June to 4 July 2009.

## 3.1 Comparison of aerosol vertical profiles

In order to assess the ability of the different retrieval algorithms to determine the general structure of the boundary layer, aerosol vertical profiles are compared to backscatter profiles from a co-located ceilometer instrument. For this comparison, it is important to consider that MAX-DOAS measurements have a relatively low information content. The number of independent pieces of information from the measurement, quantified by the degrees of freedom for signal (DFS), typically ranges between 1 and 2. An example for aerosol extinction averaging kernels, taken from the Heidelberg retrieval, is shown in Fig. 3. The averaging kernels indicate that information on the extinction profile can be retrieved only for the lowermost 2 km of the atmosphere with highest sensitivity at the ground, where the vertical resolution (quantified by the altitude where the averaging kernel of the lowermost layer is half of its surface value) amounts to $\approx 500$ m. The DFS strongly varies with visibility and is also a function of SZA (and to a smaller extent SAA), and amounts to 1.9 for the example in Fig. 3.

Since the ceilometer backscatter profiles are characterised by a much higher vertical and temporal resolution than MAX-DOAS measurements, 20 min averages of the ceilometer profiles were degraded to the sensitivity of the Heidelberg MAX-DOAS profiles according to the method described by Rodgers and Connor (2003). The degraded backscatter profile $x' = \mathbf{A} \cdot x$, with $\mathbf{A}$ being the averaging kernel matrix and $x$ the original backscatter profile in high resolution, represents the profile $x'$ that would have been retrieved with MAX-DOAS if the true profile was $x$. Note that the original equation from Rodgers and Connor (2003), $x' = x_a + \mathbf{A} \cdot (x - x_a)$, cannot be applied here since the backscatter profiles $x$ and the a priori extinction profiles $x_a$ are measured in different physical units. The ceilometer data has been averaged to the vertical grid

**AMTD**

doi:10.5194/amt-2015-358

**MAX-DOAS aerosol intercomparison**

U. Frieß et al.

of the MAX-DOAS retrieval (200 m) prior to the convolution with the averaging kernel. No or only limited overlap between outgoing beam and field of view of the telescope of the ceilometer is present in the lowermost 120 m. For this reason, ceilometer data between surface and 150 m altitude are set to a constant value equal to the signal at 150 m during the convolution process. Therefore the lowermost layer of the convolved ceilometer profiles is subject to large uncertainties if high gradients near the surface exist. It is important to note that ceilometer and MAX-DOAS instruments retrieve different quantities. The MAX-DOAS retrieval algorithms yield extinction profiles, whereas the backscatter profiles from the ceilometer cannot be directly converted to an extinction profile without further assumptions on the ratio between backscatter and extinction. This so-called lidar ratio is not known a priori and is a function of the size and optical properties of the particles, which vary with time and altitude. Therefore ceilometer and MAX-DOAS profiles can only be compared qualitatively in terms of the vertical structure of the boundary layer. Furthermore, the MAX-DOAS instruments average over a large horizontal distance of up to several tens of kilometres, whereas the ceilometer probes the atmosphere directly over the measurement site.

The MAX-DOAS extinction profiles from the different groups together with the ceilometer backscatter profiles for the golden days are shown in Figs. 4–7. Note that BIRA, Heidelberg and AIOFM retrieve the aerosol extinction on a vertical grid of 200 m, whereas JAMSTEC represents the profile on four layers of 1 km thickness each, and MPIC retrieves the height and AOT of a box profile with a constant extinction from the surface up to a certain altitude (as well as an exponentially decreasing profile above, which contains 5 % of the AOT). The gaps in the data sets are caused by different quality filters applied by the different groups, and by missing data around noon when reference measurements were performed.

In general, the vertical structure of the aerosol profile in the boundary layer of all groups shows good agreement with the ceilometer backscatter profiles, in particular after these are degraded to the MAX-DOAS vertical resolution by convolution with the averaging kernel. The temporal variation of the MAX-DOAS profiles is in good agree-

Discussion Paper | Discussion Paper | Discussion Paper | Discussion Paper

ment with the ceilometer data, and the height of the boundary layer is generally captured very well.

The 23 and 24 June are characterised by a relatively low extinction ($< 0.4\,\mathrm{km}^{-1}$), with an increase both in boundary layer height and in extinction in the early afternoon. These features are captured well by all groups. An enhanced backscatter at $\approx 1.5\,\mathrm{km}$ altitude in the early afternoon of 23 June, probably due to clouds, is captured by the retrievals of BIRA, Heidelberg, AIOFM and JAMSTEC which show uplifted layers of enhanced extinction during this period (no data is reported for this period by MPIC). However, as a consequence of the limited information content of MAX-DOAS measurements, these layers are smeared out over a layer extending from 200 m to 1.2 km. A similar situation with an uplifted aerosol layer in the early afternoon occurs on 25 June. After 06:00 UTC on 25 June, a cloud is observed by the ceilometer at an altitude of $\approx 2\,\mathrm{km}$, which is still visible after convolution with the averaging kernel. The finding that none of the MAX-DOAS retrievals captures this cloud might be due to the fact that it is localised directly over the measurement site, whereas the MAX-DOAS extinction profiles are representative for the atmosphere in a distance of several kilometres along the line of sight. In fact, the Heidelberg and AIOFM profiles exhibit layers of enhanced extinction ($\approx 0.15\,\mathrm{km}^{-1}$) between 0.5 and 2 km throughout the morning of 25 June which probably correspond to the cloud layer observed by the ceilometer in zenith between 6 and 8 a.m.

As can be seen from the webcam images in Fig. 8, foggy conditions prevailed during the mornings of 30 June and 1 July. The Ceilometer backscatter profiles show that these thin fog layers with a vertical extent in the order of 100 m were initially located very close to the surface and then uplifted during the course of the morning. Note that the backscatter profiles smoothed with the MAX-DOAS averaging kernel do not show an enhanced extinction in the early morning of 30 June because the fog layer was located at altitudes below 150 m, and was therefore not considered in the smoothing procedure. These foggy conditions allow for an investigation of the behaviour of the retrieval algorithms in the presence of a layer of high extinction at different altitudes. As shown in Fig. 9, the diurnal variation of the DFS in the presence of fog is similar

**AMTD**

doi:10.5194/amt-2015-358

**MAX-DOAS aerosol intercomparison**

U. Frieß et al.

to the clear sky case. An enhanced extinction in the morning due to fog is detected by all retrieval algorithms. However, the limited vertical resolution in the presence of fog leads to a strong overestimation of the vertical extent of the extinction layer. On 30 June, the fog layer present during the early morning hours is blurred over an al-

5 titude of 0.5 km by BIRA, MPIC and AIOFM and 1 km by Heidelberg and JAMSTEC, respectively. On 1 July, the retrieved fog layer extends up to 1.3, 1.8 and 1–2 km for the BIRA, Heidelberg and MPIC retrievals, respectively, whereas the fog layer detected by the Ceilometer was located below 500 m until 09:00 UTC. The vertical profiles retrieved by BIRA, Heidelberg and AIOFM are, however, qualitatively in good agreement

with the expected profile as given by the Ceilometer profiles smoothed with the MAX-DOAS averaging kernels. None of the algorithms is able to reproduce the elevated extinction layers occurring after the uplift of the fog layers in the course of the mornings of 30 June and 1 July. This might be caused by the general enhancement in extinction throughout the boundary layer on these two days, which could lead to a reduced sensi-

tivity for higher altitudes. This is in contrast to the situation on 25 June, when elevated layers could be detected during conditions of lower aerosol load.

Although the BIRA, Heidelberg and AIOFM algorithms are very similar in terms of the parametrisation of the aerosol profile and the choice of the a priori, the resulting profiles exhibit some differences, which can be either caused by a different choice of

20 the a priori profiles and a priori covariance matrices, or in case of BIRA in addition by a larger number of elevation angles with a higher sensitivity near the surface due to the inclusion of measurements at 1° elevation. A persistent feature of the BIRA profiles is a reduced extinction in the lowermost (0–200 m) layer with significantly smaller values than in the layers above, even if the Ceilometer indicates a homogeneous distribution in

the boundary layer (e.g., on 24 and 25 June). However, the Ceilometer does not have any information on altitudes below 150 m, and it might well be that the surface acts as a sink for aerosols or that increased relative humidity leads to larger particles and thus higher extinction at higher altitudes. On the other hand, the BIRA and AIOFM algorithms seem to be able to capture uplifted layers or clouds better than the Heidelberg

**AMTD**

doi:10.5194/amt-2015-358

**MAX-DOAS aerosol intercomparison**

U. Frieß et al.

and JAMSTEC algorithms, e.g. in the afternoon of 30 June and the midday of 1 July. Both BIRA and JAMSTEC detect an uplifted layer of enhanced extinction the morning of 2 July, when clouds were present, a feature that is not captured by the Heidelberg and AIOFM algorithm. In some cases, such as the late afternoon of 2 and 3 July, the AIOFM profiles show an enhanced extinction between 1 and 2 km altitude, where the Ceilometer also detects enhanced backscatter, probably due to clouds at the top of the boundary layer. This enhanced sensitivity for clouds at higher altitudes is probably due to the fact that the AIFOM algorithm includes relative intensities in addition to $O_4$ dSCDs in the measurement vector, which render the algorithm more sensitive to enhanced extinction at higher altitudes (Frieß et al., 2006). The AIOFM profiles exhibit a somewhat higher temporal variability, which is either due to the shorter time interval for each profile (about 7 min compared to 20 and 30 min for the other algorithms), and/or because the inclusion of relative intensities leads to a higher sensitivity to short-term variations due to clouds.

On 3 July a closed cloud cover is present between 08:30 and 14:30 UTC. The Ceilometer profiles show that the cloud base is initially located at very low altitudes (< 250 m) and increases in height in the early afternoon, leading to an uplifted layer after 12:00 UTC. These features are also present in the Ceilometer profiles degraded with the MAX-DOAS averaging kernel. Under these conditions, the BIRA, Heidelberg and AIOFM algorithms are able to retrieve the vertical structure of the boundary layer realistically, although some differences exist in the detailed structure and the height and vertical extent of the extinction layer in the afternoon. In particular, AIFOM does not detect the uplift until 13:30 UTC, but detects enhanced extinction between 1 and 2 km altitude corresponding to a thin cloud layer at the top of the boundary layer visible in the ceilometer profiles between 14:00 and 18:00 UTC. In contrast, the coarse representation of the profile by JAMSTEC and the parametrised algorithm by MPIC both show an enhancement in extinction due to the presence of clouds but are not capable of retrieving the uplifted layer in the afternoon.

Discussion Paper | Discussion Paper | Discussion Paper | Discussion Paper |

**AMTD**

doi:10.5194/amt-2015-358

**MAX-DOAS aerosol intercomparison**

U. Frieß et al.

## 3.2 Comparison of AOT and surface extinction

In this section, the AOT and surface extinction derived by the different participants is compared to sun photometer and WetNeph measurements, respectively. The AOT is either derived by integrating the extinction profile (BIRA, Heidelberg, AIOFM and JAMSTEC) or directly retrieved (MPIC). For BIRA, Heidelberg, AIOFM and JAMSTEC, the value of the lowermost retrieval layer is considered as being representative for the surface extinction, whereas for MPIC the AOT divided by the layer height serves as an estimate. The time series of AOT and surface extinction for the golden days of the CINDI campaign are shown in Fig. 10.

An overall good agreement between the AOT from MAX-DOAS and from the sun photometer is achieved. Under conditions of clear sky and low aerosol load (e.g., 23 and 24 June), BIRA tends to underestimate the AOT in the afternoon, when the relative azimuth angle between viewing direction and Sun (RAA) is small. In contrast, MPIC tends to underestimate the AOT in the morning under conditions of high aerosol load (1 and 3 July). Best agreement between all MAX-DOAS measurements as well as sun photometer is achieved under clear sky conditions in the morning hours when the RAA is large (23 and 24 June as well as 4 July). The larger differences between the different workgroups at higher and more variable aerosol load (30 June–3 July) is either caused by differences in the retrieval algorithm or by slightly different temporal and/or spatial sampling (i.e., slightly different viewing directions). As already discussed in Sect. 3.1, the very high AOT and surface extinction values observed during the morning of 30 June and 1 July are caused by fog. Unfortunately, no sun photometer measurements are available for these periods since these rely on the observation of direct sunlight. The same applies to the morning of 25 June and the noon of 3 July. However, the overall good agrement between the vertical profiles from MAX-DOAS and ceilometer (see Sect. 3.1) provide confidence that the AOT can be retrieved reliably even under these conditions of reduced visibility.

**AMTD**

doi:10.5194/amt-2015-358

**MAX-DOAS aerosol intercomparison**

U. Frieß et al.

Title Page

Abstract | Introduction

Conclusions | References

Tables | Figures

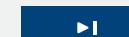

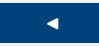 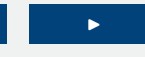

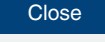

Discussion Paper | Discussion Paper | Discussion Paper | Discussion Paper |

A sudden jump in the AOT values from the sun photometer occurs on 2 July at 14:30 UTC, but is not apparent in the MAX-DOAS data. It is not clear whether this is caused by local aerosols not captured by the MAX-DOAS instrument due to the different viewing geometry, or by erroneous sun photometer data. For these reasons, the data of the afternoon of 2 July is excluded from the following correlation analysis.

The correlation between the AOT from MAX-DOAS and from sun photometer as well as histograms for the AOT difference (MAX-DOAS minus sun photometer) for the different workgroups are shown in Fig. 11. The results of the regression analyses are listed in Table 2. The correlation coefficient is $> 0.8$ for all workgroups, and the mean difference (accuracy) between AOT from MAX-DOAS and sun photometer is $< 0.07$ with a standard deviation $< 0.1$ (precision). All datasets exhibit a slope significantly smaller than one, ranging from 0.62 (MPIC) to 0.90 (JAMSTEC). This systematic underestimation of the AOT is likely to be caused by both the fact that the sensitivity for high altitudes is low and that the partial AOT above the altitude where aerosol extinction has been retrieved (4 km) has not been considered in this analysis. Best agreement in terms of slope (0.9) and mean difference to sun photometer measurements (0.01) is achieved by JAMSTEC. However, compared to the other participants the difference of JAMSTEC data to sun photometer AOT shows a large scatter (0.092), and no data has been submitted by JAMSTEC for the late afternoon (after 16:00 UTC) when the RAA is small and systematic problems with the retrieval might occur, leading to the smallest number of datapoints (73) submitted by this group.

It is important to note that parts of the discrepancies between the AOT from different workgroups does not only originate from the different retrieval strategies and parametrisations, as well as from the different time periods for which data is available from the different groups, but in case of MPIC also from the fact that the inversions are based on $O_4$ measurements at a different wavelength. The MPIC retrieval is based on measurements of the 360 nm $O_4$ absorption band, and the retrieved extinction is converted to 477 nm using the Ångström coefficient derived from co-located sun photometer measurements. Therefore likely reasons for the small slope in the AOT comparision be-

**AMTD**

doi:10.5194/amt-2015-358

**MAX-DOAS aerosol intercomparison**

U. Frieß et al.

Title Page

Abstract | Introduction

Conclusions | References

Tables | Figures

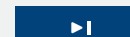

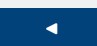 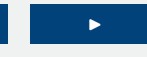

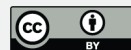

Discussion Paper | Discussion Paper | Discussion Paper | Discussion Paper

**AMTD**

doi:10.5194/amt-2015-358

**MAX-DOAS aerosol intercomparison**

U. Frieß et al.

tween MPIC and sun photometer are both the uncertainties in the Ångström coefficient and the reduced visibility in the UV which leads to a different horizontal footprint of the MPIC observations.

As shown in the upper panel of Fig. 10, a strong disagreement exists between the surface extinction of the different MAX-DOAS retrievals and the in-situ measurements from the WetNeph instrument, especially in the afternoon and during periods of high aerosol load. The WetNeph observes a much smaller extinction than all MAX-DOAS retrievals for most of the time. This is also apparent in the correlation plots and the histograms of differences between MAX-DOAS and WetNeph surface extinction shown in Fig. 12. A summary of the regression analysis for surface extinction is shown in Table 3. Note that the regression analysis yields values different to those reported by Zieger et al. (2011). This is first because different samples are compared (in the present study only data from the golden days are considered), and second because Zieger et al. (2011) applied a weighted orthogonal fit whereas here a usual linear regression has been used. Best agreement in terms of mean difference between MAX-DOAS and WetNeph is achieved by the parametrised MPIC algorithm which is not capable to directly determine gradients in the aerosol extinction near the surface. Hence a possible explanation for the strong discrepancies observed for the OEM algorithms (BIRA, Heidelberg, AIOFM and JAMSTEC) could be a strong increase in extinction below the height of the WetNeph inlet (60 m above ground). Further possible reasons for these discrepancies and a comprehensive statistical analysis of data from the CESAR site for an extended period of time have already been discussed in detail by Zieger et al. (2011). We still do no have a conclusive explanation for the origin of these differences, in particular since both the AOT and the vertical structure of the boundary layer are captured well by the MAX-DOAS vertical profiles.

Surface extinction values from the different MAX-DOAS algorithms show good agreement during conditions of low aerosol (23–26 June), but exhibit significant discrepancies at higher aerosol load (e.g., 30 June–3 July). Again, a likely reason for parts of the discrepancies in surface extinction between the different MAX-DOAS retrievals is

the fact that different parametrisations of the extinction profile are used. Since relative humidity tends to increase with altitude in the boundary layer, hygroscopic growth of aerosol particles usually leads to an increase in extinction with altitude. Moreover, gas may partition to aerosol as RH increases and temperature reduces with increasing altitude, and ammonium and nitrate were observed to increase with altitude in the vicinity of Cabauw (Morgan et al., 2010; Aan de Brugh et al., 2012). An inhomogeneous vertical distribution leads to erroneous estimates of the surface extinction for models with a coarse vertical grid (JAMSTEC) or with parametrised retrievals (MPIC). JAMSTEC represents the extinction profile on a 1 km vertical grid and should for these reasons tend to overestimate the surface extinction if extinction increases with altitude. The same should be true for MPIC, for which the surface extinction (or rather the average boundary layer extinction) is estimated by dividing the AOT by the retrieved layer height. Indeed, JAMSTEC retrieves the highest AOTs, whereas MPIC retrieves a smaller extinction than the other workgroups under conditions of high aerosol load and large vertical gradients (30 June to 3 July). Although Heidelberg, BIRA and AIOFM use the same vertical grid with a layer thickness of 200 m and comparable retrieval algorithms, surface extinction values from these groups show significant discrepancies in cases of high aerosol load or fog, e.g. in the morning of 30 June, on 2 July and in the afternoon of 4 July.

In summary, possible reasons for the observed discrepancies between surface extinction from MAX-DOAS and WetNeph are (1) strong vertical gradients of the aerosol extinction with increased extinction below the height of the WetNeph inlet, (2) problems of the MAX-DOAS retrieval algorithms in the presence of non-homogeneous horizontal distributions (although these are not very likely given the smooth temporal variations of the MAX-DOAS and in situ data), (3) the over-estimation of the surface extinction by MAX-DOAS in the presence of lofted layers, as well as (4) inlet losses of the in-situ instruments. Note that the extinction profiles estimated from a co-located Raman LIDAR instrument agreed much better to the in-situ WetNeph values, although only a limited number of profiles could be compared, and a Mie closure showed the consistency of

**AMTD**

doi:10.5194/amt-2015-358

**MAX-DOAS aerosol intercomparison**

U. Frieß et al.

all major aerosol in-situ measurements in the basement of the CESAR tower (Zieger et al., 2011).

## 4  Conclusions

We have presented a first direct intercomparison of aerosol extinction profiles, AOT and surface extinction from MAX-DOAS measurements. MAX-DOAS data collected during the CINDI campaign have been compared to independent measurements of the AOT from an Aeronet sun photometer, of the vertical structure from a commercial ceilometer instrument, and of the surface extinction from in situ instruments.

The retrieval algorithms which were part of this study follow very different approaches, and use different parametrisations of the aerosol vertical profiles. BIRA, Heidelberg, AIOFM and JAMSTEC use the optimal estimation method and retrieve the extinction profiles at different altitude grids (BIRA, AIOFM and Heidelberg: 200 m layers; JAMSTEC: 1 km layers). MPIC uses a least squares algorithm with the AOT and layer height as retrieval parameters, and use no further a priori constraints.

Despite large conceptual differences between the algorithms and different representations of the aerosol extinction profile, and although the information content of the MAX-DOAS measurements is low (typically in the order of two degrees of freedom for signal), the comparison of the retrieved profiles with ceilometer backscatter profiles shows that all algorithms are able to provide an estimate for the vertical extent of the boundary layer with the expected accuracy. BIRA, AIOFM and Heidelberg with the finest vertical grid of 200 m, but also to a certain extent JAMSTEC with a 1 km vertical grid, are able to resolve the vertical structure of the boundary layer and to detect uplifted aerosol layers, fog and clouds in the lowermost $\approx 1.5$ km of the atmosphere. The vertical resolution is, however, limited by the small information content of the measurements and amounts to $\approx 500$ m at the surface and $\approx 1$ km at 1 km altitude. Therefore, thin layers of high extinction, such as fog, appear strongly blurred in the retrieved extinction profiles. Unfortunately, the AOT retrieved under conditions of low visibility is

Discussion Paper | Discussion Paper | Discussion Paper | Discussion Paper

**AMTD**

doi:10.5194/amt-2015-358

**MAX-DOAS aerosol intercomparison**

U. Frieß et al.

**AMTD**

doi:10.5194/amt-2015-358

**MAX-DOAS aerosol intercomparison**

U. Frieß et al.

difficult to validate since sun photometer measurements, which rely on direct sunlight, are not available for these periods.

In general, the time series of AOT retrieved from MAX-DOAS shows good agreement with co-located sun photometer measurements. A regression analysis shows correlation coefficients better than 0.8 for all groups. All retrieval algorithms systematically underestimate the AOT with slopes of ranging from 0.6 to 0.9, and mean AOT differences (MAX-DOAS minus sun photometer) of less than 0.07. It is important to note that parts of the differences between MAX-DOAS and sun photometer are probably due to the fact that both kinds of instruments observe different air masses in a highly populated and polluted region where horizontal gradients in aerosol load are likely to occur. Furthermore, MAX-DOAS is insensitive to aerosols above $\approx 2\,\text{km}$. In case of MPIC, additional systematic differences might be caused by the conversion of the AOT from 360 to 477 nm.

Given that the AOT and the vertical structure of the extinction profiles are captured reasonably well by the different retrieval algorithms, it remains open why there is such a large discrepancy between the surface extinction from MAX-DOAS and from Wet-Neph. In particular in the afternoon, the WetNeph shows much smaller values than retrieved by MAX-DOAS. Significant differences between the individual MAX-DOAS retrievals, in particular under conditions of high aerosol load and large vertical gradients, can be partially explained by the different parametrisations of the vertical profile. Furthermore, strong vertical gradients in aerosol extinction near the surface are a potential reason for the observed discrepancies.

Although the ability of MAX-DOAS measurements to determine vertical profiles of aerosols is limited by a small information content and a relatively low vertical resolution, this intercomparison study shows that the MAX-DOAS technique can reliably determine the vertical structure of the lowermost 2 km of the atmosphere. For typical Mid-European conditions as observed during the CINDI campaign, the AOT can be retrieved with an accuracy better than 0.07 and a precision better than 0.09, and ob-

Discussion Paper | Discussion Paper | Discussion Paper | Discussion Paper |

servations are not limited to clear sky conditions but can also be performed during situations of low visibility.

*Acknowledgements.* The CINDI Campaign was for a large part funded by the ESA project CEOS Intercalibration of ground-based spectrometers and lidars (ESRIN contract 22202/09/I-EC) and the EU project ACCENT-AT2 (GOCE-CT-2004-505337). We further acknowledge the support of the EU via the GEOMon Integrated Project (contract FP6-2005-Global-4-036677). The work of Frieß and Yilmaz has been financially supported by the EU Project EUSAAR (contract 2006-026140). Many thanks to Ankie Piters, Jennifer Hains and Mark Kroon for hosting the campaign and for the superb organisation. The excellent logistical support by Jacques Warmer and the staff of KNMI at the CESAR site is gratefully acknowledged. Many thanks to Alexei Rozanov from IUP Bremen for providing the SCIATRAN radiative transfer model.

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

**Table 1.** Main features of the different retrieval algorithms.

| Participant | Method[*] | Measurement | Wavelength | Retrieved Quantities | Vertical Grid | Sampling interval |
|---|---|---|---|---|---|---|
| AIOFM | OEM | $O_4$ and intensity | 477 nm | Extinction profile | 200 m | ≈ 7 min |
| BIRA | OEM | $O_4$ dSCDs | 477 nm | Extinction profile | 200 m | ≈ 20 min |
| Heidelberg | OEM | $O_4$ dSCDs | 477 nm | Extinction profile | 200 m | 20 min |
| JAMSTEC | OEM | $O_4$ dSCDs | 477 nm | Extinction profile | 1000 m | 30 min |
| MPIC | LSM | $O_4$ dSCDs | 360 nm | Layer height and AOT | n/a | ≈ 15 min |

[*] OEM: Optimal Estimation Method; LSM: Least Squares Method.

# AMTD

doi:10.5194/amt-2015-358

**MAX-DOAS aerosol intercomparison**

U. Frieß et al.

**Table 2.** Comparison between the AOT from MAX-DOAS and from sun photometer. Listed are the number of datapoints, intercept and slope of the linear regression, the correlation coefficient $R$, the mean difference (MAX-DOAS minus sun photometer) and the standard deviation of the mean difference.

| Participant | $N$ | Intercept | Slope | $R$ | ΔAOT |
|---|---|---|---|---|---|
| AIOFM | 431 | $0.071 \pm 0.007$ | $0.795 \pm 0.023$ | 0.86 | $0.011 \pm 0.079$ |
| BIRA | 140 | $0.021 \pm 0.014$ | $0.702 \pm 0.045$ | 0.80 | $-0.062 \pm 0.083$ |
| Heidelberg | 149 | $0.027 \pm 0.012$ | $0.805 \pm 0.037$ | 0.87 | $-0.031 \pm 0.078$ |
| JAMSTEC | 73 | $0.040 \pm 0.022$ | $0.902 \pm 0.062$ | 0.86 | $0.010 \pm 0.092$ |
| MPIC | 128 | $0.039 \pm 0.014$ | $0.622 \pm 0.040$ | 0.81 | $-0.071 \pm 0.100$ |

**AMTD**

doi:10.5194/amt-2015-358

**MAX-DOAS aerosol intercomparison**

U. Frieß et al.

Discussion Paper | Discussion Paper | Discussion Paper | Discussion Paper |

**Table 3.** Comparison between the surface extinction from MAX-DOAS and from WetNeph. Listed are the number of datapoints, intercept and slope of the linear regression, the correlation coefficient $R$, the mean difference (MAX-DOAS minus WetNeph) and the standard deviation of the mean difference. All extinction values are in units of $km^{-1}$.

| Participant | $N$ | Intercept | Slope | $R$ | ΔAOT |
|---|---|---|---|---|---|
| AIOFM | 617 | $-0.025 \pm 0.017$ | $3.773 \pm 0.200$ | 0.61 | $0.165 \pm 0.298$ |
| BIRA | 180 | $0.014 \pm 0.006$ | $1.638 \pm 0.115$ | 0.73 | $0.096 \pm 0.122$ |
| Heidelberg | 215 | $0.023 \pm 0.007$ | $2.328 \pm 0.086$ | 0.88 | $0.105 \pm 0.099$ |
| JAMSTEC | 112 | $0.046 \pm 0.008$ | $1.214 \pm 0.144$ | 0.63 | $0.132 \pm 0.103$ |
| MPIC | 158 | $0.025 \pm 0.011$ | $1.492 \pm 0.099$ | 0.77 | $0.070 \pm 0.076$ |

**AMTD**

doi:10.5194/amt-2015-358

**MAX-DOAS aerosol intercomparison**

U. Frieß et al.

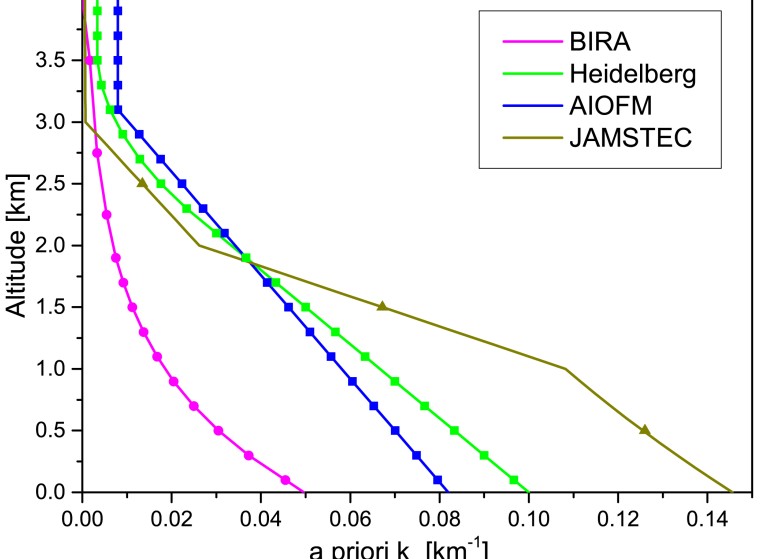

**Figure 1.** A priori profiles for the BIRA, Heidelberg, AIOFM and JAMSTEC retrievals. The symbols indicate the centre of each retrieval layer. The BIRA, Heidelberg and AIOFM algorithm use a 200 m vertical grid with constant extinction in each layer, and JAMSTEC a 1 km grid with exponentially decreasing extinction in each layer.

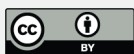

Discussion Paper | Discussion Paper | Discussion Paper | Discussion Paper |

**AMTD**

doi:10.5194/amt-2015-358

**MAX-DOAS aerosol intercomparison**

U. Frieß et al.

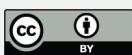

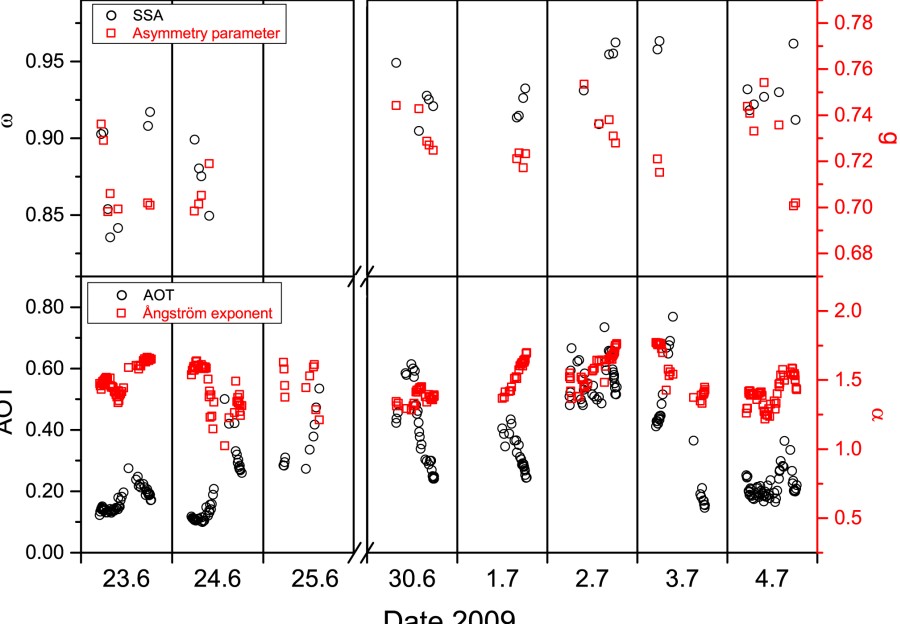

**Figure 2.** Summary of the Aeronet data obtained during the golden days of the CINDI campaign. Top: single scattering albedo $\omega$ and asymmetry parameter $g$ at 441 nm from almucantar measurements; bottom: AOT at 440 nm and Ångström coefficient $\alpha$ retrieved from direct sunlight measurements at 440 and 675 nm.

**AMTD**

doi:10.5194/amt-2015-358

**MAX-DOAS aerosol intercomparison**

U. Frieß et al.

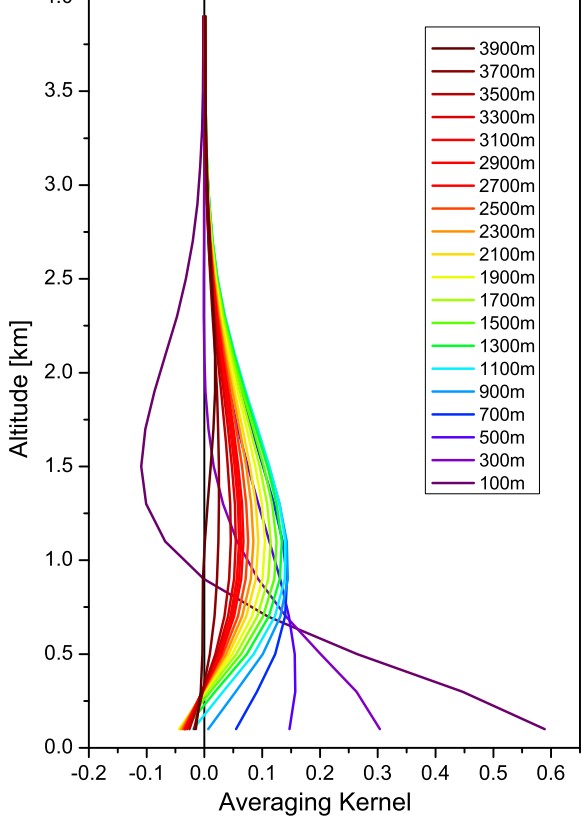

**Figure 3.** Example for aerosol extinction averaging kernels from the Heidelberg retrieval algorithm for 2 July 2009, 12:00 UTC.

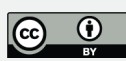

**Figure 4.** Comparison of MAX-DOAS extinction profiles with backscatter profiles from the ceilometer for 23 June (left) 24 June (right). The top panel shows the backscatter signal in original vertical resolution, the second panel the backscatter signal with the averaging kernels of Heidelberg applied, below the extinction profiles retrieved from BIRA, Heidelberg, JAMSTEC and MPIC. For MPIC box-profiles with the retrieved layer height and AOT are plotted.

**AMTD**

doi:10.5194/amt-2015-358

**MAX-DOAS aerosol intercomparison**

U. Frieß et al.

**AMTD**

doi:10.5194/amt-2015-358

**MAX-DOAS aerosol intercomparison**

U. Frieß et al.

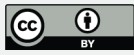

Figure 5. Same as Fig. 5, but for 25 June (left) and 30 June (right).

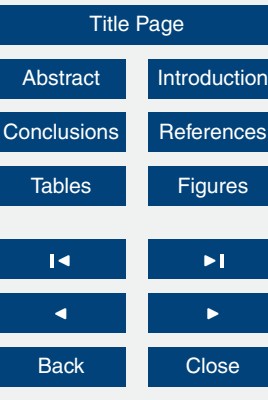

**Figure 6.** Same as Fig. 6, but for 1 July (left) and 2 July (right).

**AMTD**

doi:10.5194/amt-2015-358

**MAX-DOAS aerosol intercomparison**

U. Frieß et al.



**AMTD**

doi:10.5194/amt-2015-358

**MAX-DOAS aerosol intercomparison**

U. Frieß et al.

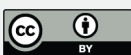

**Figure 7.** Same as Fig. 7, but for 3 July (left) 4 July (right).

Discussion Paper | Discussion Paper | Discussion Paper | Discussion Paper

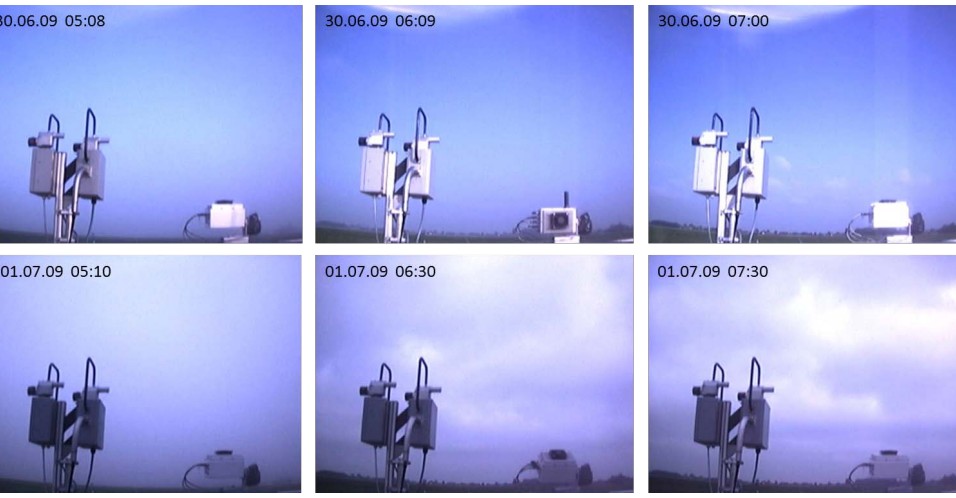

**Figure 8.** Webcam images from the mornings of 30 June and 1 July 2009, when foggy conditions prevailed. The webcam pointed to the viewing direction of the MAX-DOAS Instruments (north-east). The devices in the foreground are the telescopes of the Heidelberg group.

**AMTD**

doi:10.5194/amt-2015-358

**MAX-DOAS aerosol intercomparison**

U. Frieß et al.

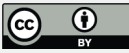

Discussion Paper | Discussion Paper | Discussion Paper | Discussion Paper | Discussion Paper |

# AMTD

doi:10.5194/amt-2015-358

**MAX-DOAS aerosol intercomparison**

U. Frieß et al.

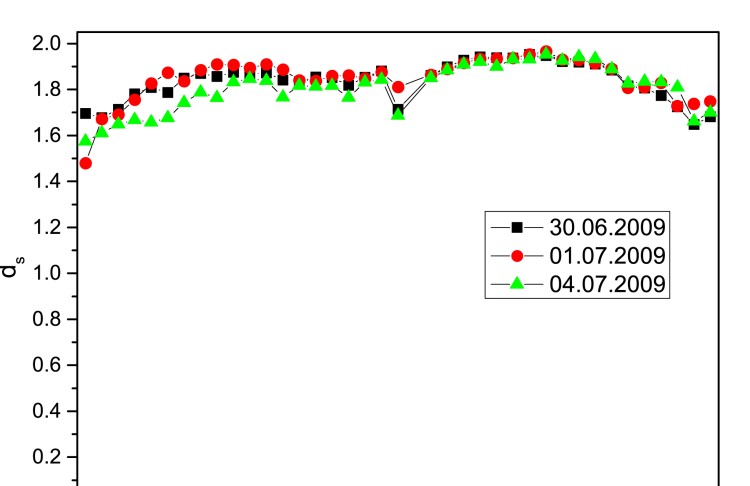

**Figure 9.** Diurnal variation of the degrees of freedom for signal from the Heidelberg retrieval for 30 June as well as 1 and 4 July 2009.

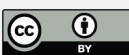

**AMTD**

doi:10.5194/amt-2015-358

**MAX-DOAS aerosol intercomparison**

U. Frieß et al.

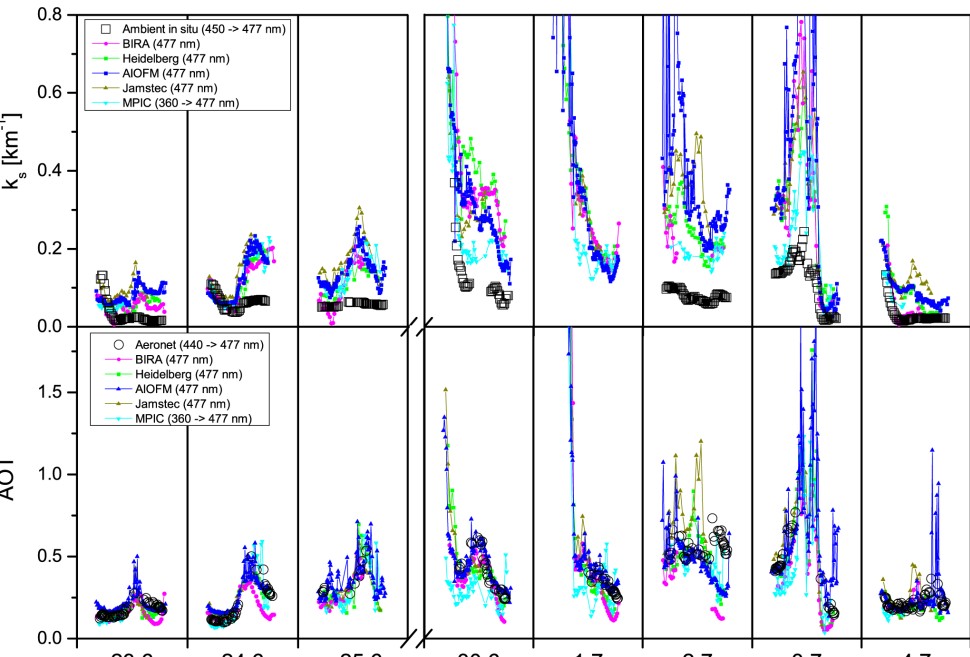

**Figure 10.** Time series of particle light extinction coefficient determined at the ground level (top) and AOT (bottom) from the MAX-DOAS retrieval (coloured symbols), together with the surface extinction from the humidity controlled nephelometer (open squares) and the AOT from the sun photometer (open circles) for the golden days of the CINDI campaign. All data is converted to a wavelength of 477 nm using the Ångström coefficient derived from sun photometer measurements.

**AMTD**

doi:10.5194/amt-2015-358

MAX-DOAS aerosol intercomparison

U. Frieß et al.

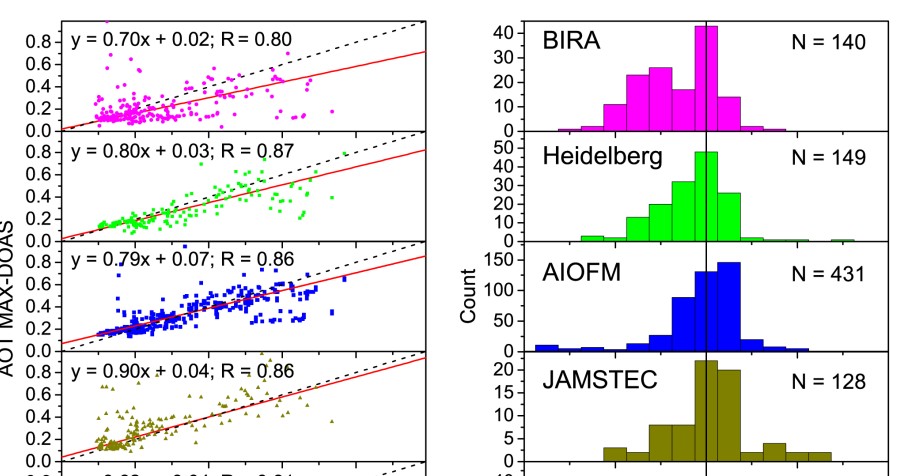

**Figure 11.** Left panels: correlation between AOT from the different workgroups and from the sun photometer. The red line shows the linear fit and the dashed line the 1 : 1 line. Right panels: histograms of the difference in AOT (MAX-DOAS – sun photometer).

Discussion Paper | Discussion Paper | Discussion Paper | Discussion Paper

**AMTD**

doi:10.5194/amt-2015-358

**MAX-DOAS aerosol intercomparison**

U. Frieß et al.

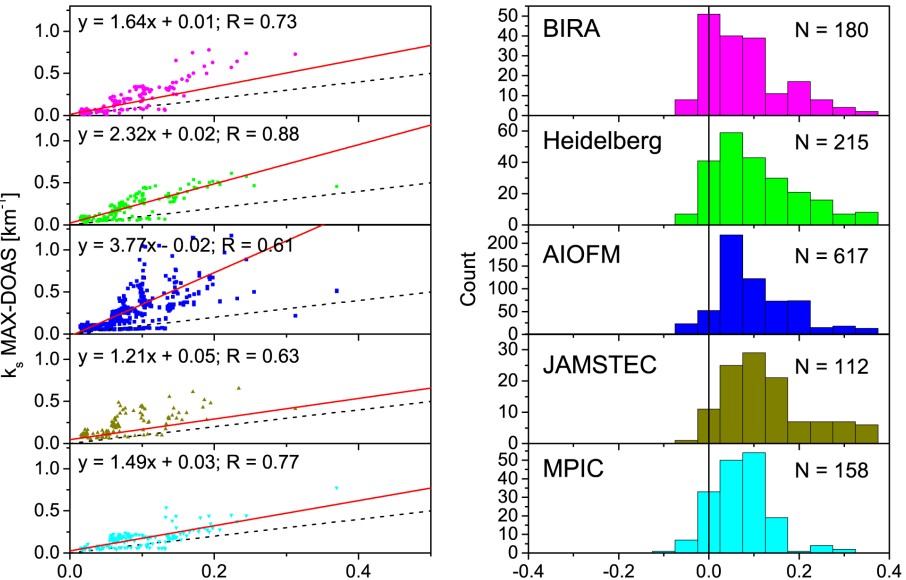

**Figure 12.** Left panels: correlation between surface extinction from the different workgroups and from WetNeph. The red line shows the linear fit and the dashed line the 1 : 1 line. Right panels: histograms of the difference in surface extinction (MAX-DOAS – WetNeph).