# Peer review of "Intercomparison of aerosol extinction profiles retrieved from MAX-DOAS measurements"

_Atmospheric Measurement Techniques, 2015_

## Referee Comment (RC1) · Anonymous Referee #1 · 15 Mar 2016

Frieß et al present inter-comparison of AOT and surface extinction coefficients retrieved during CINDI 2009 by five different groups (algorithms) from MAX-DOAS data, AERONET and humidity controlled nephelometer. They also compare aerosol extinction coefficient profiles from MAX-DOAS and ceilometer backscatter profiles smoothed on MAX-DOAS measurements sensitivity vertical and temporal grid below 4 km. Most algorithms use optimal estimation iterative fitting of the measured and modeled dSCD of the oxygen collision complex (O2O2). SSA, asymmetry parameter (Henyey–Greenstein approximation), and surface reflectivity are input parameters that are derived from external sources. The main conclusions of the paper: 1) MAX-DOAS AEC profiles are relatively well captured while comparing with the smoothed ceilometer backscatter profiles that have no sensitivity below 150 m; 2) relatively good correlation (R ≈ 0.8) with the AERONET AOT but with the systematic underestimation of 10 –

30% by MAX-DOAS; 3) Surface MAX-DOAS AEC is significantly larger than the in-situ nephelometer AEC at 60 m. Good discussion of the MAX-DOAS validation difficulties is given and potential causes of disagrements. The paper is well written, clearly presents methods and results. The topic is relevant to AMT and I strongly recommend publishing the article after some modifications.

Major concern:

In my opinion the paper should address in more detail the "correction" factor of 0.8 +/- 0.1 used to decrease the observed dSCD(O2O2) to match the modeled dSCD(O2O2). This correction factor is mentioned in description of MCIP and AIOFM algorithms, but based on the previous publications, it is applied by all participating groups. Originally thought to be caused by the T-dependence of O2O2 absorption cross section, it is not supported by direct sun and airborne MAX-DOAS (Spinei et al., 2015) measurements. Recent study by Ortega et al., 2016 suggests that increase in dSCD(O2O2) is due to elevated aerosol layers. If this is the case dSCD(O2O2) have larger sensitivity to the aerosol elevated layers than it is commonly assumed and the application of the correction factor is not acceptable for aerosol retrieval. Underestimation of the MAX-DOAS derived AOT relative to AERONET is of about the same magnitude as the dSCD correction factor.

Minor comments:

Section 2.4:

What is the source of MPIC temperature, pressure and relative humidity profiles during CINDI? Using soundings launched over deBilt gives maximum VCD(O2O2) of 1.32 molecules2/cm5 vs 1.43 molecules2/cm5.

Section 3.1

Averaging kernels are the result of the OE retrieval so different averaging kernels will be produced by different groups depending on their algorithm implementation and input parameters. Since the a priori is not a true climatology, the same MAX-DOAS measurements will have seemingly different vertical sensitivities. Non-OE algorithms have no easy way to generate averaging kernels. I find it somewhat misleading to show ceilometer data convolved with the Heidelberg averaging kernels as to "what to expect" for MAX-DOAS retrievals from all groups. How do the authors define the PBL height from the MAX-DOAS aerosol profiles? Figures 4-7 show rather large variability between the groups in vertical distribution of AEC.

In my opinion it will be useful to add "lessons learned" section to elaborate on the potential improvement of MAX-DOAS aerosol validation during CINDI-2016 campaign (e.g. needed in-situ and remote sensing instrumentation, observation geometries, etc.)

Table 2 and 3:

Please add "*" to JAMSTEC data and an explanation below: * Only data points at UTC before 16:00 are reported

Figure 2. SSA and asymmetry factors from AERONET CIMEL are shown only for parts of some days. How do the groups estimate these inputs when there are no AERONET retrievals?

Figure 6. July 2 panels are not aligned with July 1 2009 panels.

Figure 7. Why the gap in ceilometer data smoothed by Heidelberg averaging kernels during noon reference (zenith) measurements is narrower than in the Heidelberg retrieval data that produced the averaging kernels?

Figure 10. It is difficult to see individual group results. I suggest having a panel with the "reference" data and then plot the differences to the reference data (maybe in percent to the reference?)

References:

Ortega, I., Berg, L. K., Ferrare, R. A., Hair, J. W., Hostetler, C. A. and Volkamer,

R.: Elevated aerosol layers modify the O2–O2 absorption measured by ground-based MAX-DOAS, Journal of Quantitative Spectroscopy and Radiative Transfer, 176, 34–49, doi:10.1016/j.jqsrt.2016.02.021, 2016.

Spinei, E., Cede, A., Herman, J., Mount, G. H., Eloranta, E., Morley, B., Baidar, S., Dix, B., Ortega, I., Koenig, T. and Volkamer, R.: Ground-based direct-sun DOAS and airborne MAX-DOAS measurements of the collision-induced oxygen complex, O2O2, absorption with significant pressure and temperature differences, Atmospheric Measurement Techniques, 8(2), 793–809, doi:10.5194/amt-8-793-2015, 2015.

---

## Referee Comment (RC2) · Anonymous Referee #2 · 15 Apr 2016

The authors present an intercomparison of aerosol measurements made during the CINDI campaign held in summer 2009 with emphasis on MAX-DOAS retrievals of aerosol profiles, in particular aerosol extinction profiles and AOT as retrieved from O4 DSCDs. The actual MAX-DOAS retrievals for the different groups are only discussed briefly within this manuscript but sufficient reference material is provided covering the retrieval methods in more detail.

The MAX-DOAS aerosol data sets are then compared with independent aerosol measurements made during the CINDI campaign as well and the following conclusions were drawn by the authors. First, the MAX-DOAS aerosol profiles are compared to smoothed backscatter profiles from a ceilometer and show good agreement regarding the vertical structure of aerosol in the boundary layer. Second, the MAX-DOAS AOT is compared with the AOT from an Aeronet sun photometer with the MAX-DOAS AOT

time series showing overall good agreement with the sun photometer measurements but all MAX-DOAS retrievals systematically underestimate the AOT with potential reasons being briefly discussed in the manuscript. Third, substantial differences exist between the MAX-DOAS surface aerosol extinction when compared to the data measured with the in-situ nephelometer. Potential reasons are discussed but it is acknowledged that the clear disagreement remains largely unresolved.

I have no major comments that need to be addressed. The manuscript is well written, the content is presented clearly, and the paper is recommended for publication in AMT.

Comments to be addressed before publication:

Page 4, line 14: Delete 'a' before 'relatively'

Page 4, line 15: Replace 'On the other hand,' with something like 'Furthermore'

Page 4, line 17: Add one sentence to say if MAX-DOAS can address the issues mention above or not, then continue with 'The usage . . .'

Page 5, lines 1,2: Change to (or rather add) 'Compared to lidar profiles, MAX-DOAS measurements have . . .'

Page 5, line 10: Could add: '... FOV of the receiving telescope of the MAX-DOAS instrument . . .'

Page 5, line 4 and other places: My preference is to rather use 'group' instead of 'workgroup' as used on page 22, line 25.

Page 9, line 13: This should read: '. . .CINDI campaign with their own instrument, but used data . . . '

Page 9, lines 15-29: Wouldn't it make sense for every group to agree and then use the same or as similar as possible a priori?

Page 12, line 23: replace 'Our . . .' with 'The MPIC retrieval . . .'

[Figure]

Page 15, line 1: 'golden days' should be explained where first used, currently explained on page16, lines 2-3.

Page 16, lines 19-22: Please explain briefly if and/or how smoothing the ceilometer profiles with the Heidelberg average kernels might have impacted on the comparison study, i.e. how different would the smoothed ceilometer profiles have looked if the averaging kernels of a different group would have been used.

Page 18, line 21: Use 'ceilometer' (no capital c, not consistent within the manuscript).

Page 19, lines 17-22: Sounds somewhat contradictory: the authors write that BIRA, Heidelberg and AIOFM are similar re their choice in a priori but then they point out that the difference between them could also be caused by the different choice of a priori. That needs some rewording or clarification. And it raises again the question if this could be avoided by streamlining the a priori used for the retrievals as much as possible between the groups.

Page 19, lines 20-22: In case of the BIRA retrievals, these should be redone using the some subset of elevation angles the other groups used which - when compared with the original BIRA set (including all the angles in the retrieval) - would then show if this causes some of the differences seen in the profiles or not.

Page 20 & Figure 7: Somewhere in the discussion should also be mentioned that the AIOFM retrieval gets the elevated cloud layer in the afternoon of 4 July right – actually a very nice example and rather impressive. However, that is not at all the case in the Heidelberg data set which uses the same MAX-DOAS data, correct? Any explanation? Interestingly, the elevated layer is not visible in the ceilometer data set smoothed with the Heidelberg averaging kernels either – any thoughts??? Maybe I missed that but didn't see any discussion in the text.

Page 21, line 25: typo 'agreement'

Page 23, line 23L typo 'do not have'

---

## Author Comment (AC1) · 11 May 2016

We thank the anonymous reviewer for the constructive comments, which are very helpful for an improvement of our manuscript. In the following, reviewer comments are cited in *italic*.

*Frieß et al present inter-comparison of AOT and surface extinction coefficients retrieved during CINDI 2009 by five different groups (algorithms) from MAX-DOAS data, AERONET and humidity controlled nephelometer. They also compare aerosol extinction coefficient profiles from MAX-DOAS and ceilometer backscatter profiles smoothed on MAX-DOAS measurements sensitivity vertical and temporal grid below 4 km. Most algorithms use optimal estimation iterative fitting of the measured and modeled dSCD of the oxygen collision complex (O2O2). SSA, asymmetry parameter (Henyey- Green-*

[Figure]

*stein approximation), and surface reflectivity are input parameters that are derived from external sources. The main conclusions of the paper: 1) MAX-DOAS AEC profiles are relatively well captured while comparing with the smoothed ceilometer backscatter profiles that have no sensitivity below 150 m; 2) relatively good correlation (R ? 0.8) with the AERONET AOT but with the systematic underestimation of 10 - C1 30*

*Major concern:*

*In my opinion the paper should address in more detail the "correction" factor of $0.8 \pm 0.1$ used to decrease the observed dSCD(O2O2) to match the modeled dSCD(O2O2). This correction factor is mentioned in description of MCIP and AIOFM algorithms, but based on the previous publications, it is applied by all participating groups.*

A scaling factor for the measured $O_4$ dSCDs is indeed applied by all groups. This correction factor is not a focus of the present study since the potential necessity to apply such a factor, its influence of the agreement of simulated and measured $O_4$ dSCDs and the accuracy of the resulting extinction profiles and AOT, as well as possible causes of the observed discrepancies when not applying this correction have been addressed by several other publications (e.g., Clèmer et al,. 2010, Spinei et al., 2015, and Ortega et al., 2016). We will add the following statement to the conclusions section: "A further source of uncertainty is the empirical correction factor for the $O_4$ dSCDs, for which a value of 1.2 - 1.3 has been applied by all participating groups. This correction factor has not been the focus of the present paper, but recent studies indicate that the disagreement between modelled and measured $O_4$ dSCDs is probably not caused by uncertainties in the temperature dependence of the $O_4$ cross section, but that elevated aerosol layers might be a potential cause."

*Originally thought to be caused by the T-dependence of O2O2 absorption cross section, it is not supported by direct sun and airborne MAX-DOAS (Spinei et al., 2015) measurements. Recent study by Ortega et al., 2016 suggests that increase in dSCD(O2O2) is due to elevated aerosol layers. If this is the case dSCD(O2O2)*

*have larger sensitivity to the aerosol elevated layers than it is commonly assumed and the application of the correction factor is not acceptable for aerosol retrieval. Underestimation of the MAXDOAS derived AOT relative to AERONET is of about the same magnitude as the dSCD correction factor.*

In accordance with the findings from the recent studies mentioned by the reviewer, we agree that the disagreement between modelled and measured $O_4$ dSCDs is most probably not due to a temperature dependence of the cross section. However, the remaining disagreement of about 20% between MAX-DOAS and sun photometer AOD is certainly not caused by the correction factor since the retrieved AOD would be even smaller without correction (see Clèmer et al,. 2010), and the discrepancy between MAX-DOAS and sun photometer would further increase.

*Minor comments: Section 2.4: What is the source of MPIC temperature, pressure and relative humidity profiles during CINDI? Using soundings launched over deBilt gives maximum VCD(O2O2) of 1.32 molecules2/cm5 vs 1.43 molecules2/cm5.*

We agree with the reviewer that the $O_4$ VCD assumed by MPIC is too high. Since the retrieved $O_4$ DSCDs are converted into DAMFs by dividing them by the atmospheric $O_4$ VCD in the MPIC algorithm, a smaller $O_4$ VCD is equivalent with a larger $O_4$ correction factor. Therefore we will state an $O_4$ VCD of $1.32 \cdot 10^{43}$ molec$^2$/cm$^5$ and a conversion factor of 1.3 in the revised version of the manuscript.

*Section 3.1 Averaging kernels are the result of the OE retrieval so different averaging kernels will be produced by different groups depending on their algorithm implementation and input parameters. Since the a priori is not a true climatology, the same MAX-DOAS measurements will have seemingly different vertical sensitivities. Non-OE algorithms have no easy way to generate averaging kernels. I find it somewhat misleading to show ceilometer data convolved with the Heidelberg averaging kernels as to "what to expect" for MAX-DOAS retrievals from all groups.*

We agree that the averaging kernels depend on the choice of the a priori, not only

because the a priori is not a true climatology. However, the limited vertical resolution represented by the averaging kernel is mainly determined by the limited information content of the measurements and not by the a priori (see e.g. Frieß et al., 2006). Therefore, we can expect that the resulting convoluted ceilometer profiles using averaging kernels from the different groups are very similar.

*How do the authors define the PBL height from the MAX-DOAS aerosol profiles? Figures 4-7 show rather large variability between the groups in vertical distribution of AEC.*

We do not provide a quantitative definition of the PBL height, and a common definition would be difficult due to the differences in the parametrisation of the vertical profiles by the different algorithms. It is obvious from Figures 4-7 that significant differences exist between the data sets, and we hope that the discussion of the discrepancies in the structure and height of the extinction profiles are discussed sufficiently in section 3.1.

*In my opinion it will be useful to add "lessons learned" section to elaborate on the potential improvement of MAX-DOAS aerosol validation during CINDI-2016 campaign (e.g. needed in-situ and remote sensing instrumentation, observation geometries, etc.)*

This point is quite difficult to answer since there was already a large and almost complete suite of aerosol instrumentation present during CINDI-I, and similar measurements will be performed during CINDI-II. Thus not the amount of data available for comparison is the limiting factor, but rather the amount of aerosol parameters retrieved from the MAX-DOAS measurements. Specifically, I think about the retrieval of aerosol optical and microphysical properties (single scattering albedo, phase function, size distribution and complex refractive index), which are preferably retrieved from azimuthal scans which will most probably be performed by several groups during CINDI-II. However, at the current state I am not aware of any algorithm available for the retrieval of such parameters (although the sensitivity to aerosol optical parameters has been demonstrated by Frieß et al., 2006), and I am therefore hesitant to give specific recommendations regarding these aspects at the current stage.

*Table 2 and 3: Please add "*" to JAMSTEC data and an explanation below: * Only data points at UTC before 16:00 are reported*

An according footnote will be added to the revised manuscript.

*Figure 2. SSA and asymmetry factors from AERONET CIMEL are shown only for parts of some days. How do the groups estimate these inputs when there are no AERONET retrievals?*

Indeed, aerosol optical and microphysical properties from AERONET are only available from almucantar measurements every few hours during clear sky conditions. However, the variability of these parameters, in particular of the asymmetry parameter, are quite small. Therefore we expect that a temporal interpolation of these parameters that we have applied introduces only small errors compared to other error sources.

*Figure 6. July 2 panels are not aligned with July 1 2009 panels.*

This will be corrected in the revised version of the manuscript.

*Figure 7. Why the gap in ceilometer data smoothed by Heidelberg averaging kernels during noon reference (zenith) measurements is narrower than in the Heidelberg retrieval data that produced the averaging kernels?*

Thank you for pointing this out. There was an error in the script that performs the convolution of the ceilometer profiles which led to a wrong assignment of the averaging kernel to the respective time interval at the beginning of a data gap. We have recalculated the convoluted profiles and the figures in the revised version will contain correct data.

*Figure 10. It is difficult to see individual group results. I suggest having a panel with the "reference" data and then plot the differences to the reference data (maybe in percent to the reference?)*

We appreciate this suggestion. However, data from MAX-DOAS covers a larger time

range than Sun photometer data (which is only available during clear sky). Thus we would omit a large fraction of the data in the absence of Sun photometer data if we would plot the difference between MAX-DOAS and Sun photometer AOT.

---

## Author Comment (AC2) · 11 May 2016

We thank the anonymous reviewer for the constructive comments, which are very helpful for an improvement of our manuscript. In the following, reviewer comments are cited in *italic*.

*The authors present an intercomparison of aerosol measurements made during the CINDI campaign held in summer 2009 with emphasis on MAX-DOAS retrievals of aerosol profiles, in particular aerosol extinction profiles and AOT as retrieved from O4 DSCDs. The actual MAX-DOAS retrievals for the different groups are only discussed briefly within this manuscript but sufficient reference material is provided covering the retrieval methods in more detail.*

*The MAX-DOAS aerosol data sets are then compared with independent aerosol mea-*

[Figure]

*surements made during the CINDI campaign as well and the following conclusions were drawn by the authors. First, the MAX-DOAS aerosol profiles are compared to smoothed backscatter profiles from a ceilometer and show good agreement regarding the vertical structure of aerosol in the boundary layer. Second, the MAX-DOAS AOT is compared with the AOT from an Aeronet sun photometer with the MAX-DOAS AOT time series showing overall good agreement with the sun photometer measurements but all MAX-DOAS retrievals systematically underestimate the AOT with potential reasons being briefly discussed in the manuscript. Third, substantial differences exist between the MAX-DOAS surface aerosol extinction when compared to the data measured with the in-situ nephelometer. Potential reasons are discussed but it is acknowledged that the clear disagreement remains largely unresolved.*

*I have no major comments that need to be addressed. The manuscript is well written, the content is presented clearly, and the paper is recommended for publication in AMT.*

*Comments to be addressed before publication:*

*Page 4, line 14: Delete 'a' before 'relatively'*

This will be corrected in the revised manuscript.

*Page 4, line 15: Replace 'On the other hand,' with something like 'Furthermore'*

This will be replaced.

*Page 4, line 17: Add one sentence to say if MAX-DOAS can address the issues mention above or not, then continue with 'The usage . . .'*

We will add the sentence "Multi-Axis Differential Optical Absorption Spectroscopy (MAX-DOAS) measurements allow for the retrieval of aerosol extinction profiles, and to a certain extent also aerosol microphysical and optical properties, in the planetary boundary layer.".

*Page 5, lines 1,2: Change to (or rather add) 'Compared to lidar profiles, MAX-DOAS*

*measurements have . . .'*

The word "profiles" will be added.

*Page 5, line 10: Could add: '... FOV of the receiving telescope of the MAX-DOAS instrument . . .'*

The receiving telescope is part of the Lidar instrument, not the MAX-DOAS.

*Page 5, line 4 and other places: My preference is to rather use 'group' instead of 'workgroup' as used on page 22, line 25.*

The term "workgroup" will be changed to "group" in the revised manuscript

*Page 9, line 13: This should read: '. . .CINDI campaign with their own instrument, but used data . . . '*

This will be changed.

*Page 9, lines 15-29: Wouldn't it make sense for every group to agree and then use the same or as similar as possible a priori?*

The general approach of this study was the intercomparison of extinction profiles from what the individual groups consider as 'best' settings for their retrieval. Therefore no common a priori was defined.

*Page 12, line 23: replace 'Our . . .' with 'The MPIC retrieval . . .'*

This will be replaced.

*Page 15, line 1: 'golden days' should be explained where first used, currently explained on page16, lines 2-3.*

In the revised manuscript, we will state that "golden days" refer to days with predominantly clear sky conditions on the first occurrence of this term.

*Page 16, lines 19-22: Please explain briefly if and/or how smoothing the ceilometer*

*profiles with the Heidelberg average kernels might have impacted on the comparison study, i.e. how different would the smoothed ceilometer profiles have looked if the averaging kernels of a different group would have been used.*

The averaging kernels indeed depend on the choice of the averaging kernel, not only because the a priori is not a true climatology. However, the limited vertical resolution represented by the averaging kernel is mainly determined by the limited information content of the measurements and not by the a priori (see e.g. Frieß et al., 2006). Therefore, we can expect that the resulting convoluted ceilometer profiles using averaging kernels from the different groups are very similar.

*Page 18, line 21: Use 'ceilometer' (no capital c, not consistent within the manuscript).*

This will be corrected in the revised manuscript.

*Page 19, lines 17-22: Sounds somewhat contradictory: the authors write that BIRA, Heidelberg and AIOFM are similar re their choice in a priori but then they point out that the difference between them could also be caused by the different choice of a priori. That needs some rewording or clarification. And it raises again the question if this could be avoided by streamlining the a priori used for the retrievals as much as possible between the groups.*

We agree that these statements are contradictory. Furthermore, the a priori of BIRA significantly differs from the a priori of the other groups. We will therefore delete the statement that the a priori profiles are "very similar".

*Page 19, lines 20-22: In case of the BIRA retrievals, these should be redone using the some subset of elevation angles the other groups used which - when compared with the original BIRA set (including all the angles in the retrieval) - would then show if this causes some of the differences seen in the profiles or not.*

Due to the variety in the retrieval approaches from the different groups, the general approach of the study was the comparison of the retrieval algorithms "as they are", i.e.

without posing any limitations on the retrieval settings and the input parameters.

*Page 20 Figure 7: Somewhere in the discussion should also be mentioned that the AIOFM retrieval gets the elevated cloud layer in the afternoon of 4 July right – actually a very nice example and rather impressive. However, that is not at all the case in the Heidelberg data set which uses the same MAX-DOAS data, correct? Any explanation? Interestingly, the elevated layer is not visible in the ceilometer data set smoothed with the Heidelberg averaging kernels either – any thoughts??? Maybe I missed that but didn't see any discussion in the text.*

We will add the following sentence to the revised manuscript: "The clouds apparent in the ceilometer profiles in the afternoon of July 4 between 15:30 - 18:00 UTC are identified in the extinction profiles retrieved by the AIOFM algorithm, but not in the Heidelberg data (no other groups reported profiles for this period)."

*Page 21, line 25: typo 'agreement' Page 23, line 23L typo 'do not have'*

The typos will be corrected

―――――――――――――――